# Efficient Source-free Unlearning via Energy-Guided Data Synthesis and Discrimination-Aware Multitask Optimization

Xiuyuan Wang [1]  Chaochao Chen [* 1]  Weiming Liu [1]  Xinting Liao [1]  Fan Wang [1]  Xiaolin Zheng [1]

## Abstract

With growing privacy concerns and the enforcement of data protection regulations, machine unlearning has emerged as a promising approach for removing the influence of forget data while maintaining model performance on retain data. However, most existing unlearning methods require access to the original training data, which is often impractical due to privacy policies, storage constraints, and other limitations. This gives rise to the challenging task of source-free unlearning, where unlearning must be accomplished without accessing the original training data. Few existing source-free unlearning methods rely on knowledge distillation and model retraining, which impose substantial computational costs. In this work, we propose the Data Synthesis-based Discrimination-Aware (DSDA) unlearning framework, which enables efficient source-free unlearning in two stages: (1) Accelerated Energy-Guided Data Synthesis (AEGDS), which employs Langevin dynamics to model the training data distribution while integrating Runge–Kutta methods and momentum to enhance efficiency. (2) Discrimination-Aware Multitask Optimization (DAMO), which refines the feature distribution of retain data and mitigates the gradient conflicts among multiple unlearning objectives. Extensive experiments on three benchmark datasets demonstrate that DSDA outperforms existing unlearning methods, validating its effectiveness and efficiency in source-free unlearning.

## 1. Introduction

Modern Machine Learning (ML) models rely on vast amounts of data for training, which may contain sensitive or private information, posing significant privacy risks (Cao & Yang, 2015; Nguyen et al., 2022; Liu et al., 2023a;b). To mitigate these risks, multiple regulations, e.g., the European Union's General Data Protection Regulation (GDPR) (Voigt & Von dem Bussche, 2017) and the California Consumer Privacy Act (CCPA) (Pardau, 2018), mandate companies and organizations to implement data deletion mechanisms and grant individuals the right to be forgotten. This gives rise to the field of *machine unlearning*, whose primary goal is to ensure that the model eliminates all influence of the data requested for deletion (i.e., forget data) while preserving model integrity and performance on the remaining data (i.e., retain data).

Existing methods typically assume access to the training data, utilizing gradient computation or weight adjustment techniques to erase specific knowledge. Some methods estimate the influence of training data on model parameters, using the Fisher Information Matrix (FIM) (Foster et al., 2024; Golatkar et al., 2020) and Hessian (Mehta et al., 2022), which are prohibitively expensive due to the high dimensionality of the parameter space. Other methods retrain (Bourtoule et al., 2021; Chundawat et al., 2023a) or fine-tune (Tarun et al., 2023) the model with the training data, intentionally degrading model performance on forgot data while preserving its performance on retain data.

However, access to the original training data cannot be guaranteed in practical scenarios, due to privacy concerns, data retention policies, or other constraints. For example, many cloud platforms delete training data immediately after use to address privacy concerns and storage limitations. Similarly, in streaming data environments, real-time processing overwrites old data with new inputs, preventing historical data retention. Under these circumstances, inaccessible training data makes most existing unlearning methods infeasible. Consequently, there is a pressing need for unlearning without the training data, relying solely on the original model and limited auxiliary information (class labels) to perform unlearning, referred to as **source-free unlearning**.

Source-free unlearning is an emerging yet underexplored area, with only a few existing methods attempting to tackle its challenges. Specifically, GKT (Chundawat et al., 2023b) and ISPF (Zhang et al., 2024) both adopt the Data-Free

---

[1]Zhejiang University, China. Correspondence to: Chaochao Chen <zjuccc@zju.edu.cn>.

*Proceedings of the 42$^{nd}$ International Conference on Machine Learning*, Vancouver, Canada. PMLR 267, 2025. Copyright 2025 by the author(s).

Knowledge Distillation (DFKD) technique with a filtering mechanism to selectively transfer knowledge from the original model to a randomly initialized model. However, they both require training a new model from scratch during the distillation process, resulting in significant computational costs.

Motivated by the limitations of existing methods, we propose the Data Synthesis-based Discrimination-Aware (**DSDA**) unlearning framework to achieve efficient and effective source-free unlearning, which consists of the following two key stages:

(i) In the first stage, we overcome the unavailability of training data by proposing the *accelerated energy-guided data synthesis (AEGDS)* method to generate synthetic datasets. Specifically, we derive an energy function by reinterpreting the output logits of the original model and employ Langevin dynamics to implicitly model the training data distribution. To further improve efficiency, we incorporate high-order Runge–Kutta methods and momentum-based updates into the sampling process, reducing redundant sampling steps while preserving effectiveness.

(ii) In the second stage, we propose the *Discrimination-Aware Multitask Optimization (DAMO)* method, which utilizes the synthetic datasets for effective unlearning. Through feature space visualization, we observe that traditional unlearning losses disrupt feature distributions, causing retain class samples to become widely dispersed, leading to performance degradation. In light of this, we introduce a novel unlearning objective, *discriminative feature alignment objective*, to improve intra-class compactness and inter-class separability of retain classes, thereby improving model performance. Additionally, to resolve gradient conflicts arising from optimizing the triple objectives, i.e., forget, retain, and feature alignment, we develop a multitask optimization strategy, ensuring stability and balance in unlearning.

Our main contributions are summarized as follows:

• We propose DSDA, a novel two-stage framework for efficient, source-free unlearning, addressing the limitations of methods that rely on original data or incur high computational costs.

• For the first stage, we propose AEGDS to efficiently generate synthetic datasets as substitutes for the original data. For the second stage, motivated by insights from feature space analysis, we propose DAMO, an unlearning optimization method that enhances feature distribution and resolves gradient conflicts.

• Extensive experiments on three benchmark datasets demonstrate DSDA's superiority over existing methods across multiple tasks.

## 2. Related Work

### 2.1. Deep Machine Unlearning

Existing deep unlearning methods can be categorized based on their dependency on training data.

**Methods Requiring Full Training Data.** Many existing methods depend on access to the complete training data. Exact unlearning methods (Bourtoule et al., 2021; Yan et al., 2022; Kim & Woo, 2022) remove the forget data from the original dataset and retrain the model, incurring high computational costs. Approximate unlearning methods bypass complete retraining, with some methods leveraging Hessian matrices (Sekhari et al., 2021; Mehta et al., 2022; Li et al., 2023b) or Fisher Information Matrices (FIM) (Foster et al., 2024; Golatkar et al., 2020) to estimate and reverse the influence of forget data on model parameters, while others (Chundawat et al., 2023a; Thudi et al., 2022; Chen et al., 2024; Li et al., 2023a) fine-tune the original model directly using the training data.

**Methods Requiring Partial Training Data.** When retain data is unavailable, Cha et al. (2024); Kim et al. (2024) leverage Projected Gradient Descent (PGD) (Mądry et al., 2017) to generate adversarial samples for the forget classes. SCAR Bonato et al. (2025) substitutes external datasets for the original data, and performs unlearning via Knowledge Distillation (KD). When forget data is unavailable, UNSIR (Tarun et al., 2023) generates noise matrices for the forget classes with an error-maximization mechanism to induce class-level unlearning. However, all these methods rely on full or partial access to training data, which is often infeasible in real-world scenarios.

**Source-Free Unlearning.** In the strictest setting of source-free unlearning, where neither the forget nor retain data is accessible, there are only few existing methods. GKT (Chundawat et al., 2023b) represents the first source-free unlearning method, applying DFKD within an adversarial inversion-and-distillation framework. Building on GKT, ISFP (Zhang et al., 2024) addresses the over-filtering issue in DFKD by introducing inhibited synthesis to reduce the generation of forgetting-related information. However, both methods rely on DFKD, which necessitates costly and time-consuming model retraining, limiting their practicality for large-scale or real-time unlearning. In this work, we propose the DSDA framework, which overcomes these limitations by leveraging energy-guided data synthesis and efficient fine-tuning through discrimination-aware multitask optimization.

### 2.2. Model Inversion

Model inversion (MI) aims to reconstruct training data by exploiting the model's outputs, gradients, or internal representations (Mahendran & Vedaldi, 2015). Gradient-based MI methods optimize synthetic data using techniques such

as Momentum SGD or Adam (Kingma & Ba, 2014), minimizing the loss between model predictions and ground truth (Struppek et al., 2022; Yuan et al., 2023). Alternatively, GMI (Zhang et al., 2020) employs Generative Adversarial Networks (GANs) (Goodfellow et al., 2014) to guide data generation, with subsequent works (Yuan et al., 2023; Nguyen et al., 2023; Chen et al., 2021) enhancing generator performance by integrating additional information from the target model. However, these methods are computationally expensive, requiring either frequent gradient evaluations or training of an auxiliary generator. In contrast, the proposed AEGDS manipulates the model's likelihood landscape through implicit distribution modeling (Chen et al., 2021), achieving efficient model inversion for source-free unlearning.

## 3. Method

In this section, we define the problem of source-free unlearning and introduce two key stages of the DSDA framework, as illustrated in Figure 1. In the first stage, we propose the AEGDS, a method that efficiently generates synthetic datasets as substitutes for the original training data. In the second stage, we propose the DAMO method to perform unlearning using the synthetic datasets. DAMO incorporates a novel discriminative feature alignment objective and resolves gradient conflicts from optimizing multiple unlearning objectives simultaneously through multitask optimization, ensuring stable and balanced unlearning.

### 3.1. Preliminaries and Notations

First, we formulate machine unlearning problem as follows. Let $\mathcal{D}_{train} = \{(x_{train}, y_{train})\} \in \mathcal{X} \times \mathcal{Y}$ denote the training dataset, where $\mathcal{X}$ and $\mathcal{Y} = \{1, 2, ..., K\}$ are input and class label space, respectively. Let $\mathcal{C}_f$ denote the set of classes we intend to forget in a pre-trained ML model and $\mathcal{D}_f$ denote the subset of training data corresponding to the forget classes. Similarly, $\mathcal{C}_r$ denote the set of classes we intend to retain, with the corresponding data subset $\mathcal{D}_r$, satisfying $\mathcal{D}_r = \mathcal{D}_{train} \backslash \mathcal{D}_f$. A ML model, represented by $M(\cdot; \theta)$, generates classification probabilities for each class, where $\theta$ is the set of model parameters. $M$ can be decomposed into two components: $M^f$, which represents the collection of feature extraction layers, and $M^c$, which represents the final classification layer. Let $M(\cdot; \theta_o)$ denote the *original model* trained on the complete dataset $\mathcal{D}_{train}$. In this paper, we address a challenging yet practical scenario of **source-free unlearning**, where the unlearned model is generated directly from $\theta_o$ and class information in $\mathcal{C}_f$, without requiring access to $\mathcal{D}_{train}$.

### 3.2. Accelerated Energy-Guided Data Synthesis

Since the original training data is inaccessible, we first introduce an energy-guided mechanism to generate synthetic datasets that approximate the distributions of the original data. To further improve efficiency, we propose the **AEGDS** mechanism, which integrates high-order numerical methods with momentum-based updates to reduce computational overhead while preserving the integrity of the generated samples.

**Formalizing the Energy Function**   Energy-based models (EBMs) define a probability distribution over data samples using an energy function $E(x)$, where the likelihood of a sample is inversely proportional to its energy (Grathwohl et al., 2019; LeCun et al., 2006). For an input $x \in \mathcal{X}$, the energy function $E : \mathcal{X} \rightarrow \mathbf{R}$ maps each data sample into an energy value, which can be interpreted as an unnormalized probability (Du & Mordatch, 2019). To approximate the original data distribution for each class, the energy-based model can be constructed from the original $M(\cdot, \theta_o)$ based on the observation that discriminative models inherently follow an energy-based framework (Grathwohl et al., 2019). Specifically, the energy function is derived from the logits of the classifier, defined as:

$$E_\theta(x, y) = -\log M(x, \theta_o)[y], \qquad (1)$$

where $M(x, \theta_o)[y]$ is the predicted probability of the sample $x$ belonging to class $y$.

**Energy-Guided Data Synthesis (EGDS)**   By specifying an energy function $E_\theta(x, y)$ for a target class $y$, we can then perform data synthesis following the principles of Stochastic Gradient Langevin Dynamics (SGLD) (Welling & Teh, 2011), which generates samples that align with the energy-based model by iterative gradient-guided updates combined with noise perturbations.

**Theorem 3.1.** *Given the initial data probability $q(\boldsymbol{x})$ and the target probability $p(\boldsymbol{x})$, one can determine the transformation process via $\boldsymbol{T}(\boldsymbol{x}) = \boldsymbol{x} + \boldsymbol{\epsilon}$ by minimizing KL-Divergence as shown:*

$$\begin{aligned} \min J_{\text{Langevin}} &= \text{KL}\left(q_{\boldsymbol{T}}(\boldsymbol{x}) \| p(\boldsymbol{x})\right) \\ \text{where} : \boldsymbol{T}(\boldsymbol{x}) &= \boldsymbol{x} + \boldsymbol{\epsilon} \end{aligned} \qquad (2)$$

*where $\boldsymbol{\epsilon}$ denotes the searching direction and the optimal solution on $\boldsymbol{\epsilon}$ is given as $\boldsymbol{\epsilon} = -\alpha \nabla_{\boldsymbol{\epsilon}} \left[\text{KL}\left(q_{\boldsymbol{T}}(\boldsymbol{x}) \| p(\boldsymbol{x})\right)\right]$ where $\alpha$ denotes the step size on gradient descend. The result can be further calculated as $\boldsymbol{T}(\boldsymbol{x}) = \boldsymbol{x} + \alpha \mathbb{E}_{\boldsymbol{x} \sim q(\boldsymbol{x})} \left[\nabla_{\boldsymbol{x}} \log p(\boldsymbol{x})\right].$*

*Proof.* By taking the differentiation on $\text{KL}\left(q_{\boldsymbol{T}}(\boldsymbol{x}) \| p(\boldsymbol{x})\right)$

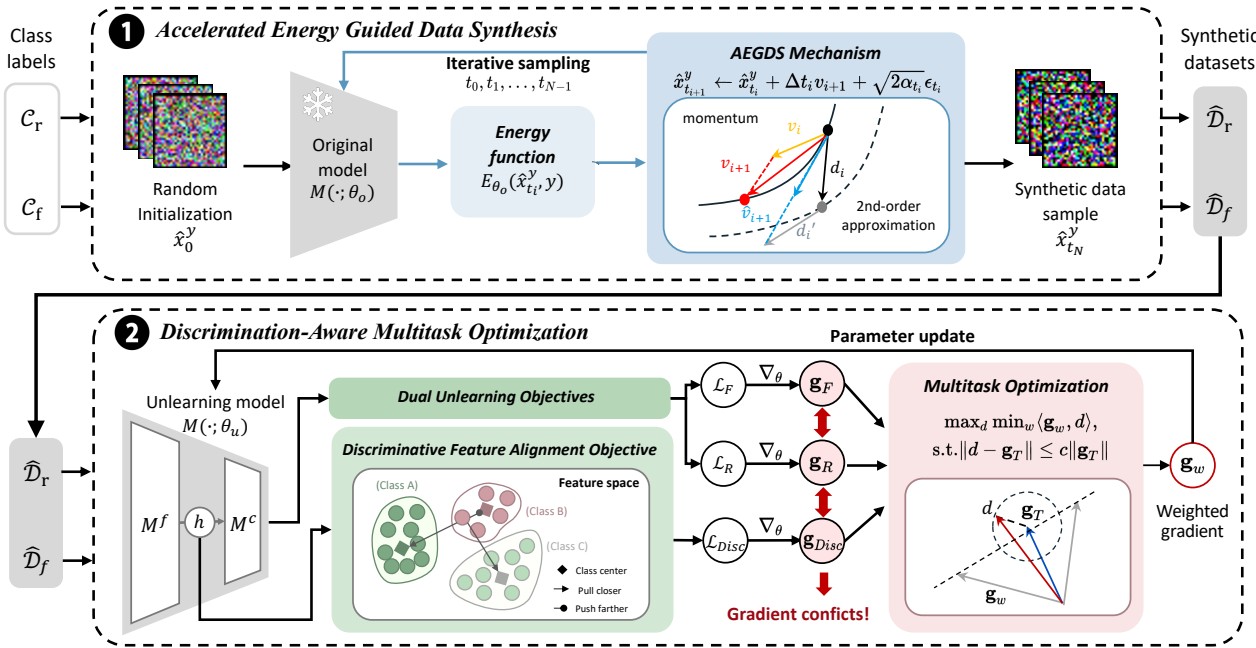

Figure 1. **Overview of the DSDA framework**, comprising two stages: (1) Given the original model and forget class labels, construct an energy function and generate synthetic datasets through AEGDS. (2) Synthetic datasets are then used to perform unlearning, incorporating dual unlearning objectives and a novel discriminative feature alignment objective, with multitask optimization for model updates.

w.r.t $\epsilon$, we can obtain the following results:

$$\nabla_\epsilon \left[ \mathrm{KL} \left( q_{\boldsymbol{T}}(\boldsymbol{x}) \| p(\boldsymbol{x}) \right) \right] = \nabla_\epsilon \left[ \mathrm{KL} \left( q(\boldsymbol{x}) \| p_{\boldsymbol{T}^{-1}}(\boldsymbol{x}) \right) \right]$$
$$= -\mathbb{E}_{\boldsymbol{x} \sim q(\boldsymbol{x})} \left[ \nabla_\epsilon \log p(\boldsymbol{T}(\boldsymbol{x})) | \det \nabla_{\boldsymbol{x}} \boldsymbol{T}(\boldsymbol{x}) | \right]$$
$$= -\mathbb{E}_{\boldsymbol{x} \sim q(\boldsymbol{x})} \left[ \nabla_\epsilon \log p(\boldsymbol{T}(\boldsymbol{x})) + \nabla_\epsilon \log | \det \nabla_{\boldsymbol{x}} \boldsymbol{T}(\boldsymbol{x}) | \right].$$

For the first term on Langevin Dynamics, it can be further calculated $\nabla_\epsilon \log p(\boldsymbol{T}(\boldsymbol{x})) = \nabla_{\boldsymbol{x}} \log p(\boldsymbol{T}(\boldsymbol{x})) \nabla_\epsilon \boldsymbol{T}(\boldsymbol{x})$ and $\nabla_\epsilon \boldsymbol{T}(\boldsymbol{x}) = \boldsymbol{I}$. Therefore, by considering the searching direction $\epsilon$ with minimizing $\mathrm{KL} \left( q_{\boldsymbol{T}}(\boldsymbol{x}) \| p(\boldsymbol{x}) \right)$, we can conclude the result as: $\boldsymbol{T}(\boldsymbol{x}) = \boldsymbol{x} - \nabla_\epsilon \left[ \mathrm{KL} \left( q_{\boldsymbol{T}}(\boldsymbol{x}) \| p(\boldsymbol{x}) \right) \right] = \boldsymbol{x} + \alpha \mathbb{E}_{\boldsymbol{x} \sim q(\boldsymbol{x})} \left[ \nabla_{\boldsymbol{x}} \log p(\boldsymbol{x}) \right]$.

Inspired by the valuable insights into Langevin dynamics presented in Theorem 3.1, we propose a novel energy-guided data synthesis mechanism, termed the **EGDS** mechanism. Specifically, we randomly initialize the synthetic data $\hat{x}_0^y$ for a target class $y$ and then update them iteratively according to the principles of SGLD, where the sampling process is carried out by constructing a Markov chain:

$$\hat{x}_{t+1}^y = \hat{x}_t^y - \alpha_t \nabla_x E_\theta(\hat{x}_t^y, y) + \sqrt{2\alpha_t} \epsilon_t, \quad (3)$$

where $x_t^y$ and $\epsilon_t$ are the sample and Gaussian noise at iteration time step $t$, $\{\alpha_i\}_{t=1}^N$ is the sequence of step size. As demonstrated in (Welling & Teh, 2011), by appropriately introducing noise and gradually reducing the step size, this procedure will converge to the distribution defined by the energy function. The pseudocode are shown in Algorithm 1

**Accelerated Energy-Guided Data Synthesis (AEGDS)** Traditional Langevin dynamics sampling, which uses a fixed time step sequence to update data states iteratively, can be computationally expensive due to frequent gradient evaluations and noise perturbations. While this stepwise progression ensures theoretical soundness, it often results in inefficient computations due to the computational cost of performing gradient evaluations and noise perturbations at every time step. To address this limitation, we propose the **AEGDS** mechanism, which skips redundant time steps while preserving sampling accuracy.

We begin by randomly initializing the synthetic data $\hat{x}_0^y$ and selecting a sequence of time steps $\{t_i\}_{i=0}^N$, where $t_{i+1}-t_i > 0$. Starting with the previous data state $\hat{x}_{t_i}^y$ at time step $t_i$, the exact solution of $\hat{x}_{t_{i+1}}^y$ at the subsequent time step $t_{i+1}$ is given by:

$$\hat{x}_{t_{i+1}}^y = \hat{x}_{t_i}^y - \alpha_{t_i} \int_{t_i}^{t_{i+1}} \nabla_x E_\theta(\hat{x}_t^y, y) dt$$
$$= \hat{x}_{t_i}^y + \alpha_{t_i} \int_{t_i}^{t_{i+1}} s_\theta(\hat{x}_t^y, y) dt, \quad (4)$$

where $s_\theta(x, y) = -\nabla_x E_\theta(x, y)$. In summary, the process is equal to progressively moving the sample from the initial time step to $t_N$, skipping intermediate steps while still capturing the target distribution.

To numerically approximate the integral, we leverage a two-

stage second-order Runge-Kutta method, which improves accuracy by incorporating intermediate gradient evaluations, reducing the errors of first-order approximations. Specifically, the synthetic data at each time step $t_i$ is updated following:

$$x' = \hat{x}^y_{t_i} + \eta \Delta t_i d_i + \sqrt{2\alpha_{t_i}} \epsilon_{t_i},$$

$$\hat{x}^y_{t_{i+1}} = \hat{x}^y_{t_i} + \alpha_{t_i} \Delta t_i \left[ \left( 1 - \frac{1}{2\eta} \right) d_i + \frac{1}{2\eta} d'_i \right] \quad (5)$$

$$+ \sqrt{2\alpha_{t_i}} \epsilon_{t_i},$$

where $d_i = s_\theta(\hat{x}^y_{t_i}, y)$ (black arrow in Figure 1)), $d'_i = s_\theta(x', y)$ (grey arrow in Figure 1)), $\Delta t_i = t_{i+1} - t_i$ and $\eta$ is a weight parameter. We set $\eta = 1$, corresponding to Heun's second-order method (Ascher & Petzold, 1998).

To further accelerate the sampling process, we incorporate Nesterov's momentum (Nesterov, 1983) which can improve convergence rates in gradient-based optimization. The momentum $v_{i+1}$ at each time step is updated as:

$$v_{i+1} = \gamma v_i + \alpha_{t_i} \hat{v}_{i+1}, \quad (6)$$

where $\hat{v}_{i+1} = \frac{1}{2} d_i + \frac{1}{2} d'_i$ (blue arrow in Figure 1)), $v_i$ is the previous momentum (yellow arrow in Figure 1)) and $\gamma$ is a momentum decay factor. The momentum term effectively leverages past gradients to accelerate convergence and smooth the updates. We formulate the overall algorithm for AEGDS in Algorithm 1. We leverage the AEGDS mechanism to perform class-wise sampling with parallel computing, generating synthetic data for each class in $\mathcal{C}_r$ and $\mathcal{C}_f$ to form $\hat{\mathcal{D}}_r$ and $\hat{\mathcal{D}}_f$.

### 3.3. Discrimination-Aware Multitask Optimization

After Section 3.2, we have obtained synthetic retain and forget datasets $\hat{\mathcal{D}}_r$ and $\hat{\mathcal{D}}_f$ Then in this section, we propose the **DAMO** method to fulfill the unlearning task . To begin with, we observe from visualization of future space that the separability and compactness of the retain classes are disrupted during unlearning, which motivates the introduction of the discriminative feature alignment objective. To further resolve gradient conflicts among multiple objectives, we develop a multitask optimization strategy for more balanced and effective unlearning.

**Dual unlearning objectives**   Following existing studies (Golatkar et al., 2020), We decompose the unlearning process into two distinct objectives, formalized as follows: The *retain objective* ensures the model maintains performance on the retain classes. This is expressed by the retain loss function $\mathcal{L}_R(\hat{\mathcal{D}}_r; \theta) = \mathcal{L}_{CE}(M(\hat{x}_r; \theta), y_r)$, where $\hat{x}_r$ represents the synthetic data samples in $\hat{\mathcal{D}}_r$ and $y_r$ represents the corresponding retain class labels. In contrast, the *forget objective* aims to degrade the model's performance on

---

**Algorithm 1** Data synthesis with EGDS/AEGDS

**Input:** Target class $y$, score function $s_\theta$, sequence of time steps $\{t_i\}_{i=0}^N$, step size $\{\alpha_{t_i}\}_{i=0}^N$, momentum decay factor $\gamma$
1: **Initialize:** $\hat{x}^y_0 \sim \mathcal{N}(0, \mathbb{I})$, $v_0 = 0$
2: **for** $i = 0$ to $N - 1$ **do**
3:    **if** adopt EGDS **then**
4:       Sample $\epsilon_i \sim \mathcal{N}(0, \mathbb{I})$
5:       Update data: $\hat{x}^y_{i+1} \leftarrow \hat{x}^y_i + \alpha_i s_\theta(\hat{x}^y_i, y) + \sqrt{2\alpha_i} \epsilon_i$
6:    **else if** adopt AEDGS **then**
7:       Sample $\epsilon_{t_i} \sim \mathcal{N}(0, \mathbb{I})$
8:       $d_i \leftarrow s_\theta(\hat{x}^y_{t_i}, y)$
9:       $x' \leftarrow \hat{x}^y_{t_i} + \Delta t_i d_i + \sqrt{2\alpha_{t_i}} \Delta t_i \epsilon_{t_i}$
10:      $d'_i \leftarrow s_\theta(x', y)$
11:      $\hat{v}_{i+1} \leftarrow \frac{1}{2} d_i + \frac{1}{2} d'_i$
12:      Update momentum: $v_{i+1} \leftarrow \gamma v_i + \alpha_{t_i} \hat{v}_{i+1}$
13:      Update data: $\hat{x}^y_{t_{i+1}} \leftarrow \hat{x}^y_{t_i} + \Delta t_i v_{i+1} + \sqrt{2\alpha_{t_i}} \Delta t_i \epsilon_{t_i}$
14:    **end if**
15: **end for**
**Output:** Generated sample $\hat{x}^y_N$ (EDGS) or $\hat{x}^y_{t_N}$ (AEDGS)

---

the forget classes. This objective is expressed through a forget loss function $\mathcal{L}_F(\hat{\mathcal{D}}_f; \theta) = -\mathcal{L}_{CE}(M(\hat{x}_f; \theta), y_f)$, where $\hat{x}_f$ and $y_f$ represent the synthetic forget data and their class labels. The overall unlearning objective is then formulated as a weighted combination of these two objectives: $\mathcal{L}_T(\hat{\mathcal{D}}_r; \hat{\mathcal{D}}_f; \theta) = \lambda_1 \mathcal{L}_R(\hat{\mathcal{D}}_r; \theta) + \lambda_2 \mathcal{L}_F(\hat{\mathcal{D}}_f; \theta)$ , where $\lambda_1$ and $\lambda_2$ are hyper-parameters.

**Discriminative Feature Alignment Objective**   However, the above objectives exhibit notable limitations when directly optimized. Specifically, we fine-tune the original model using $\mathcal{L}_T$ to achieve unlearning. By comparing the feature distributions of fine-tuned ResNet18 model on CIFAR-10 (shown in Figure 2 (c)) with those of the original and retrained ResNet18 models (shown in Figure 2 (a) and (b)), we draw two key observations: (1) In the retrained model, samples from the forgot classes are scattered near the boundary of retain classes. Moreover, compared to the original model, *the retained class samples form compact clusters with clearer separations between classes in the retrained model.* (2) Conversely, in the unlearned model, the forget class samples are more widely dispersed across multiple retained classes, which *weakens both the inter-class separability and the intra-class compactness of the retain classes.* As a result, the model's performance declines due to the disruption in feature distributions.

In light of the above observations and existing studies (Wang et al., 2025), we propose the **discriminative feature alignment objective** to improve unlearning performance. This objective is designed to simultaneously improve intra-class

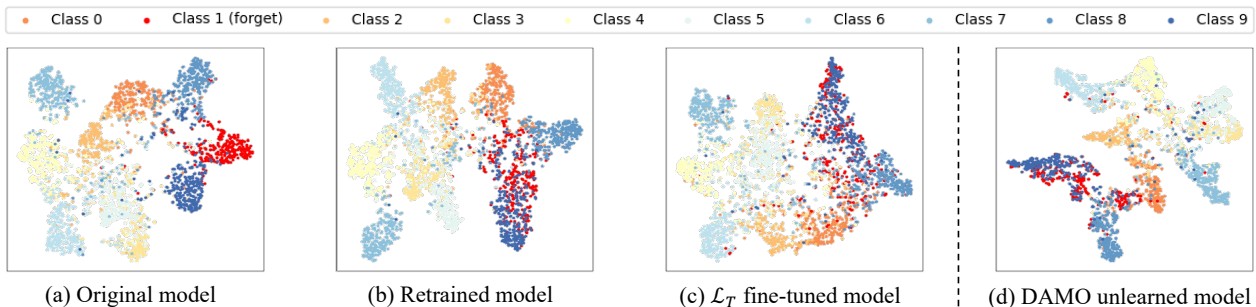

|  |  |  |  |
|---|---|---|---|
| (a) Original model | (b) Retrained model | (c) $\mathcal{L}_T$ fine-tuned model | (d) DAMO unlearned model |

*Figure 2.* Feature space visualization of the original, retrained, fine-tuned (using $\mathcal{L}_T$) and unlearned (using DAMO) ResNet18 models on the CIFAR-10 dataset. Red dots indicate the samples from $\mathcal{C}_f$, while dots in other colors represent retained samples from the $\mathcal{C}_r$.

compactness by clustering samples of the same class and enhance inter-class separability by pushing apart samples of different classes in the feature space.

Specifically, given data sample $\hat{x}_r$ belonging to class $y$, we define the inter-class similarity as $s_n^j = c_j^T h/(\|c_j\|\|h\|)$ and the intra-class similarity as $s_p = c_y^T h/(\|c_y\|\|h\|)$, where $j \in \mathcal{C}_r\backslash\{y\}$. Here, $h = M^f(\hat{x}_r; \theta)$ denotes the feature vector of the data sample, and $c_i$ denotes the feature center of class $i$. To achieve discriminative alignment, we introduce the following loss function:

$$\mathcal{L}_{Disc} = \log\left[1 + \sum_{j=1}^{|\mathcal{C}_r|-1} \exp\left(\delta\alpha_n^j s_n^j\right) \exp\left(-\delta\alpha_p s_p\right)\right], \quad (7)$$

where $\delta$ is a scale factor, $\alpha_n^j, \alpha_p$ are weight factors controlling the contribution of inter-class and intra-class similarities, respectively. To ensure stability and prevent similarity scores from deviating far from their optimal values, we define the weight factors as $\alpha_n^j = [s_n^j - O_n]_+, \alpha_p = [O_p - s_p]_+$, where $[]_+$ is the "cut-off at zero" operation that ensures non-negativity.

**Multitask Optimization for Unlearning** Optimizing the three unlearning objectives—forget, retain, and discriminative feature alignment— simultaneously is challenging due to inherent **conflicting gradients** (Yu et al., 2020). These conflicts prevent all objectives from being optimized simultaneously, leading to suboptimal performance, such as over-forgetting or under-forgetting. Let $\mathbf{g}_R = \nabla_\theta \mathcal{L}_R, \mathbf{g}_F = \nabla_\theta \mathcal{L}_F$, and $\mathbf{g}_{Disc} = \nabla_\theta \mathcal{L}_{Disc}$ be the gradients of the retain, forget, and discriminative objectives, respectively. The combined gradient is defined as $\mathbf{g}_T = \lambda_1 \mathbf{g}_R + \lambda_2 \mathbf{g}_F + \lambda_3 \mathbf{g}_{Disc}$. A gradient conflict occurs when the combined gradient $\mathbf{g}_T$ misaligns with any individual gradient, expressed as $\langle \mathbf{g}_i, \mathbf{g}_T \rangle < 0, i \in \{R, F, Disc\}$. To address this conflict, we propose a multitask optimization strategy for unlearning, which resolves gradient conflicts and balances three unlearning objectives effectively.

**Theorem 3.2.** *Given the gradients $\{\mathbf{g}_i\}, i \in \{R, F, Disc\}$, one can find an update direction $d$ that minimizes gradient conflicts while remaining close to the initial combined gradient $\mathbf{g}_T$ as:*

$$\arg\max_d \min\left\{\langle\mathbf{g}_R, d\rangle, \langle\mathbf{g}_F, d\rangle, \langle\mathbf{g}_{Disc}, d\rangle\right\}, \quad (8)$$
$$s.t. \|d - \mathbf{g}_T\| \leq c\|\mathbf{g}_T\|,$$

*where $c \in [0, 1)$ is a hyper-parameter that controls the extent of deviation from the weighted average gradient $\mathbf{g}_T$.*

To further reduce computational complexity, we introduce auxiliary weight variables $w = (w_R, w_F, w_{Disc}) \in \mathbb{R}^3$, representing the contributions of each gradient. These weights satisfy $\sum_i w_i = 1$ and $w_i \geq 0$. Equation (8) can now be reformulated as:

$$\max_d \min_w \langle\mathbf{g}_w, d\rangle, \quad s.t. \|d - \mathbf{g}_T\| \leq c\|\mathbf{g}_T\|, \quad (9)$$

where $\mathbf{g}_w = w_R \mathbf{g}_R + w_F \mathbf{g}_F + w_{Disc}\mathbf{g}_{Disc}$ represents the weighted gradient direction.

**Theorem 3.3.** *Given the optimization problem in Equation (9), one can obtain the optimal $d$ by solving the following Lagrangian:*

$$\min_{\substack{\lambda \geq 0, \\ \sum_i w_i = 1, w_i \geq 0}} \max_d \mathbf{g}_w^\top d - \frac{\lambda}{2}\|\mathbf{g}_T - d\|^2 + \frac{\lambda\phi}{2}, \quad (10)$$

*where $\phi = c^2\|\mathbf{g}_T\|^2$ represents a constraint on the update magnitude. The optimal solution can be expressed as $d^* = \mathbf{g}_T + \mathbf{g}_{w^*}/\lambda^*$, where $w^* = \arg\min_{\sum w_i = 1, w \geq 0} \mathbf{g}_w^\top \mathbf{g}_T + \sqrt{\phi}\|\mathbf{g}_w\|, \lambda^* = \|\mathbf{g}_{w^*}\|/\phi^{1/2}$.*

*Proof.* We can derive the Lagrangian of Equation (9) as:

$$\max_d \min_{\substack{\lambda \geq 0, \\ \sum_i w_i = 1, w_i \geq 0}} \mathbf{g}_w^\top d - \frac{\lambda}{2}\left(\|\mathbf{g}_T - d\|^2 - \phi\right), \quad (11)$$

where $\phi = c^2\|\mathbf{g}_T\|^2$ represents a constraint on the update magnitude. Base on Equation (11), we can derive

Equation (10) due to the concavity of the objective with respect to $d$ and the linearity of its constraints, allowing the interchange of $\max$ and $\min$ operations. Then, to solve Equation (10), we first fix $\lambda$ and $w$, and optimize the inner maximization problem. The optimal solution $d^*$ is obtained by setting the derivative with respect to $d$ to zero:

$$\frac{\partial}{\partial d}\left[\mathbf{g}_w^\top d + \lambda \mathbf{g}_T^\top d - \frac{\lambda}{2}\|d\|^2\right] = \mathbf{g}_w + \lambda \mathbf{g}_T - \lambda d = 0. \quad (12)$$

Therefore, we obtain $d^* = \mathbf{g}_T + \mathbf{g}_w/\lambda$. Substituting $d^*$ back into the objective function, the outer minimization problem simplifies to:

$$\min_{\substack{\lambda \geq 0, \\ \sum_i w_i = 1, w_i \geq 0}} \mathbf{g}_w^\top \mathbf{g}_T + \frac{\|\mathbf{g}_w\|^2}{2\lambda} - \frac{\lambda}{2}\|\mathbf{g}_T\|^2 + \frac{\lambda \phi}{2}. \quad (13)$$

Then, we compute $\lambda$ by taking the derivative and solving the resulting condition:

$$\frac{\partial}{\partial \lambda}\left(\frac{\|\mathbf{g}_w\|^2}{2\lambda} - \frac{\lambda}{2}\|\mathbf{g}_T\|^2 + \frac{\lambda \phi}{2}\right) = -\frac{\|\mathbf{g}_w\|^2}{2\lambda^2} - \frac{\|\mathbf{g}_T\|^2}{2} + \frac{\phi}{2}. \quad (14)$$

Setting the derivative to zero, the optimal $\lambda^*$ is given by $\lambda^* = \|\mathbf{g}_w\|/\sqrt{\phi}$. Substituting $\lambda^*$ back into the objective, we obtain the optimal $w^* = \arg\min_{\sum w_i = 1, w \geq 0} \mathbf{g}_w^\top \mathbf{g}_T + \sqrt{\phi}\|\mathbf{g}_w\|$, validating the correctness of Theorem 3.3.

Finally, according to Theorem 3.3, the unlearned model can be obtained by applying the optimal update direction to $\theta_o$ iteratively, which resolves gradient conflicts and maintains a balance among the unlearning objectives, thereby improving the overall unlearning performance. Additionally, to clarify the shift in feature distributions after introducing the discriminative feature alignment objective, we visualize the feature space of the unlearned model using the proposed DAMO method, as shown in Figure 2 (d). The results reveal that the retained class samples exhibit both clear inter-class separability and intra-class compactness, closely aligning with the retrained model.

## 4. Experiments

In this section, we evaluate the effectiveness of DSDA across three benchmark datasets and two model architectures. Additionally, we conduct ablation experiments to analyze the contribution of its three key components. Finally, we visualize the synthetic data to verify its integrity and privacy.

### 4.1. Experiment Settings

**Datasets and Tasks** We conduct experiments on CIFAR-10, CIFAR-100 (Krizhevsky et al., 2009) and PinsFaceRecognition (Hereis, 2024) datasets. Following existing studies (Cha et al., 2024; Foster et al., 2024), we adopt ResNet-18 (He et al., 2016) as the backbone for CIFAR-10 and CIFAR-100, and Vision Transformer (ViT) (Dosovitskiy et al., 2021) for PinsFaceRecognition.

**Baselines** We compare the proposed DSDA with the **original model** and the following unlearning methods: (1) **Retrain** refers to training a model from scratch on the retain data only. (2) **SSD** (Foster et al., 2024) selectively dampens model parameters according to their importance to the forget data, as determined by FIM. (3) **UNSIR** (Tarun et al., 2023) uses error-maximizing noise to approximate $\mathcal{D}_f$ and combines the noise with $\mathcal{D}_f$ to fine-tune the original model. (4) **ADV+IMP** (Cha et al., 2024) fine-tunes the original model using adversarial examples. (5) **LAU** (Kim et al., 2024) applys Partial-PGD and KD to the classification layer. (6) **SCAR** (Bonato et al., 2025) modifies the feature vector projections of surrogate forget data using metric learning and KD. (7) **GKT** (Chundawat et al., 2023b) employs a generator to maximize the information gap between teacher and student model, and refines model weights through gated knowledge transfer. (8) **ISPF** (Zhang et al., 2024) improves upon GKT by introducing the Inhibited Synthetic and Post-Filter methods.

**Implementation Details** We implement all experiments in Python 3.9 and use the PyTorch library (Paszke et al., 2019). All experiments are conducted on two NVIDIA RTX 3090 GPUs and repeated three times with different random seeds. Both the original and retrained models are trained from scratch using a multi-step learning rate scheduler, which begins with a learning rate of $0.01$, and optimized with the Adam optimizer (Kingma & Ba, 2014). For a fair comparison, the batch sizes of all methods are set to 256 in ResNet18 and 32 in ViT. We carefully tune all comparison methods to achieve their best performance.

**Evaluation Metrics** Following existing studies (Tarun et al., 2023; Cha et al., 2024; Foster et al., 2024), we adopt the following four metrics to measure the overall performance of an unlearning method. (1) *Accuracy on forget data ($A_f$)* evaluates the unlearned model's performance on the forget test data. (2) *Accuracy on retain data ($A_r$)* measures the unlearned model's performance on the retain test data. $A_f$ and $A_r$ of an unlearned model are expected to get close accuracy with the retrained model. (3) *Membership inference attack (MIA)* evaluates whether any information about the forget data persists in the model. We follow the logistic regression-based MIA implementation proposed in (Chundawat et al., 2023a; Foster et al., 2024). (4) *Execution time (ET)* measures the time (in seconds) required to produce the unlearned model and complete the evaluation, assessing the timeliness of the unlearning process.

### 4.2. Comparison with Baselines

To comprehensively evaluate DSDA's performance, we conduct single-class unlearning experiments across three datasets (shown in Table 1) and multi-class unlearning experiments on the CIFAR-10 dataset (shown in Table 2). Based

*Table 1.* Evaluation results for single-class unlearning. **Bolding** indicates the best result and underlining indicates the second best result.

| Method | $\mathcal{D}_r$ free | $\mathcal{D}_f$ free | CIFAR10 | | | | CIFAR100 | | | | PinsFaceRecognition | | | |
|---|---|---|---|---|---|---|---|---|---|---|---|---|---|---|
| | | | $A_r$(%) | $A_f$(%) | MIA (%) | ET ($s$) | $A_r$(%) | $A_f$(%) | MIA (%) | ET ($s$) | $A_r$(%) | $A_f$(%) | MIA (%) | ET ($s$) |
| Original | - | - | 79.80 | 88.83 | 81.40 | - | 58.66 | 45.00 | 78.66 | - | 90.85 | 88.57 | 70.09 | - |
| Retrained | - | - | 77.90 | 0.00 | 11.46 | 1108.10 | 53.98 | 0.00 | 13.22 | 2243.88 | 90.89 | 0.00 | 0.82 | 1664.66 |
| SSD | x | x | 54.44 | 0.00 | 19.72 | 168.73 | 43.59 | 0.00 | 16.80 | 160.47 | 86.76 | 0.00 | 30.00 | 598.75 |
| UNSIR | x | ✓ | 75.74 | 0.00 | 70.48 | 659.08 | 40.77 | 0.00 | 60.60 | 173.30 | 78.21 | 9.23 | 20.49 | 566.96 |
| ADV+IMP | ✓ | x | 49.30 | 0.00 | 40.68 | 143.55 | 47.39 | 2.20 | 45.42 | 180.64 | 21.01 | 0.00 | 68.11 | 731.61 |
| LAU | ✓ | x | 78.37 | 0.20 | 33.78 | **113.10** | 42.95 | 0.00 | 24.00 | **112.67** | 87.84 | 0.00 | 64.10 | **201.37** |
| SCAR | ✓ | x | 76.13 | 1.69 | 12.50 | 468.52 | 48.94 | 10.00 | 7.80 | 452.40 | 60.35 | 0.00 | 21.15 | 1248.56 |
| GKT | ✓ | ✓ | 44.23 | 2.19 | 24.98 | 393.58 | 40.74 | 0.00 | 38.38 | 461.31 | 54.14 | 2.11 | 34.87 | 1941.10 |
| ISPF | ✓ | ✓ | 66.46 | 0.00 | 31.70 | 303.97 | 45.03 | 0.00 | 23.28 | 367.50 | 55.30 | 0.00 | 32.61 | 1303.22 |
| DSDA (ours) | ✓ | ✓ | **77.91** | 0.00 | **11.80** | 133.40 | **49.96** | 0.00 | **13.80** | 155.00 | **88.36** | 0.00 | **18.39** | 549.48 |

*Table 2.* Evaluation results for multi-class unlearning on CIFAR-100. $k = |\mathcal{C}_{forget}|$ denotes the number of forget classes. **Bolding** indicates the best result and underlining indicates the second best result.

| Method | $k$=2 | | | | $k$=4 | | | | $k$=8 | | | |
|---|---|---|---|---|---|---|---|---|---|---|---|---|
| | $A_r$(%) | $A_f$(%) | MIA (%) | ET ($s$) | $A_r$(%) | $A_f$(%) | MIA (%) | ET ($s$) | $A_r$(%) | $A_f$(%) | MIA (%) | ET ($s$) |
| Original | 58.35 | 65.00 | 92.12 | - | 58.84 | 48.28 | 81.57 | - | 57.76 | 67.51 | 88.86 | - |
| Retrained | 56.23 | 0.00 | 33.21 | 1960.15 | 56.98 | 0.00 | 14.48 | 1592.25 | 57.15 | 0.00 | 31.69 | 1093.86 |
| UNSIR | 41.87 | 0.00 | 53.02 | 178.65 | 42.25 | 0.00 | 45.89 | 374.99 | 41.64 | 0.00 | 50.08 | 229.71 |
| ADV+IMP | 41.40 | 3.91 | 36.60 | 191.10 | 44.08 | 0.00 | 68.33 | 148.97 | 34.73 | 0.00 | 53.15 | 174.93 |
| LAU | 45.28 | 0.50 | 23.30 | 174.29 | 30.51 | 0.00 | 67.75 | 166.06 | 17.03 | 0.00 | 18.55 | 168.39 |
| SCAR | **52.01** | 13.50 | 28.10 | **136.79** | 42.39 | 13.50 | 34.00 | 302.61 | 40.89 | 25.00 | 35.18 | 322.40 |
| GKT | 40.57 | 0.00 | 33.65 | 568.81 | 32.74 | 0.00 | 37.95 | 895.39 | 25.39 | 0.00 | 39.38 | 973.95 |
| ISPF | 44.81 | 0.00 | 24.69 | 446.70 | 45.47 | 0.00 | 24.95 | 480.37 | 44.00 | 0.00 | 38.83 | 571.62 |
| DSDA (ours) | 49.55 | 0.00 | **33.28** | 157.25 | **50.28** | 0.00 | **17.80** | **147.01** | **49.56** | 0.00 | **28.29** | **162.55** |

on the objective of machine unlearning, we expect the performance of a desirable unlearning method to be close to the performance of the Retrain baseline. From the experimental results, we draw the following conclusions, considering both unlearning effectiveness and efficiency.

**Unlearn Effectiveness.** (1) DSDA achieves complete removal of forget data information, with $A_f$ reaching 0% across all unlearning tasks, while also attaining the highest $A_r$ among all source-free unlearning methods. Additionally, DSDA demonstrates the best or second-best accuracy performance compared to all baselines. Note that this is not a fair comparison, as source-free methods function without the original training data, making their unlearning task more challenging. (2) Several baselines, including ADV+IMP and UNSIR, exhibit significantly higher MIA values than Retrain, indicating potential leakage of forget data, despite achieving desirable accuracy. DSDA achieves the best MIA results, closely aligning with the retrained model, further demonstrating its effectiveness in mitigating privacy risks. (3) Moreover, as the number of forgotten classes increases, LAU and GKT experience notable performance degradation. In contrast, DSDA maintains robust performance regardless of the number of forget classes, underscoring its scalability and reliability in more complex unlearning scenarios.

**Unlearn Efficiency.** The ET results demonstrate that DSDA significantly outperforms source-free baselines in unlearning efficiency, with an average improvement of 68.50%.

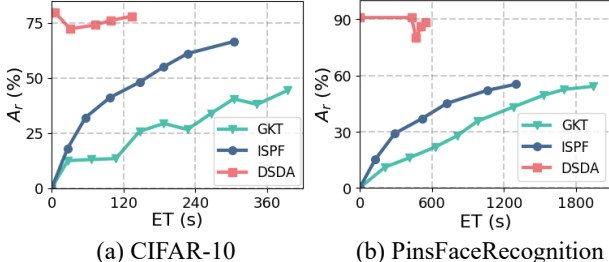

(a) CIFAR-10  (b) PinsFaceRecognition

*Figure 3.* The results of ET vs. $A_r$ under CIFAR-10 with ResNet18 and PinsFaceRecognition with ViT settings.

Furthermore, DSDA also surpasses most non-source-free baselines in efficiency, despite the additional step of generating synthetic data to substitute the original training data.

Moreover, we analyze the relationship between $A_r$ and ET for source-free unlearning methods, as shown in Figure 3. In the figure, the $A_r$ values of DSDA remain constant before a certain point, which is due to the model being frozen during the data synthesis stage. The results demonstrate that DSDA consistently achieves higher $A_r$ than GKT and ISPF, while reaching optimal performance more efficiently.

### 4.3. Ablation Study

We conduct ablation experiments on three key components in the DSDA, to elucidate their contributions respectively. Specifically, *DSDA-w-EGDS* represents DSDA without data

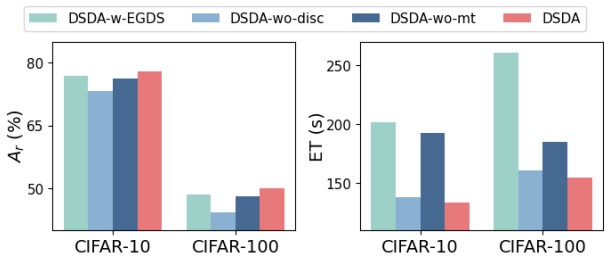

*Figure 4.* The ablation results of $A_r$ and ET on CIFAR-10 and CIFAR-100 using ResNet18.

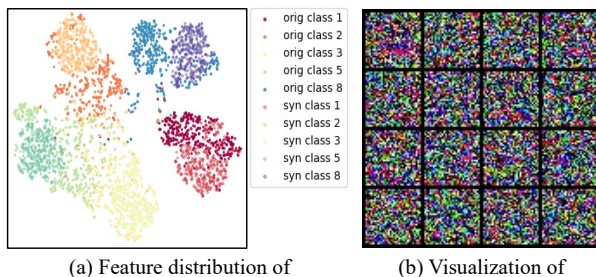

(a) Feature distribution of synthetic data and original data

(b) Visualization of synthetic data

*Figure 5.* Visualization of feature distribution and synthetic data for CIFAR-10. Round dots represent original data, square dots represent synthetic data.

synthesis acceleration. *DSDA-wo-disc* represents DSDA without the discrimination feature alignment objective, using only $\mathcal{L}_F$ and $\mathcal{L}_R$. *DSDA-wo-mt* represents DSDA without multi-task optimization, where model parameters are updated directly using $\mathbf{g}_T$.

The results of $A_r$ and ET on CIFAR-10 and CIFAR-100 datasets, as shown in Figure 4, provide several important insights: (1) DSDA-w-EGDS requires significantly longer ET than DSDA, highlighting the importance of AEGDS in improving efficiency. Additionally, DSDA outperforms DSDA-w-EGDS in $A_r$, due to the second-order Runge-Kutta method reducing approximation errors. (2) DSDA-wo-disc shows lower $A_r$ compared to DSDA, underscoring the critical role of the alignment objective in preserving intra-class compactness and inter-class separability, which in turn leads to better model performance. (3) DSDA-wo-mt leads to lower $A_r$ and higher ET, demonstrating that multi-task optimization not only balances the unlearning objectives but also accelerates the unlearning process.

### 4.4. Additional Analysis

**Feature distribution of Synthetic Data.** We visualize the feature distribution of both the synthetic and original CIFAR-10 dataset using the original model, as shown in Figure 5. The results reveal that the synthetic data closely overlaps with the original data in feature space, exhibiting nearly identical distributions, which suggests that the proposed AEGDS effectively models the original data distribution.

**Visualization of Synthetic Data.** To further evaluate the privacy implications of the synthetic data, we visualize its appearance in Figure 5 (b). The synthetic samples are visually indistinguishable, impossible for human observers to extract any meaningful information, which ensures that the synthetic data poses no privacy risks.

## 5. Conclusion

We propose DSDA, a novel source-free unlearning framework, which addresses the critical challenges of inaccessible training data and computational cost. The two key components of DSDA, i.e. AEGDS and DAMO, enable the generation of synthetic data and refinement of feature distributions, thereby ensuring both the removal of forgot knowledge and the preservation of performance on retain data. Extensive experiments on multiple benchmark datasets demonstrate that DSDA outperforms existing unlearning methods in terms of both efficiency and effectiveness. While DSDA is effective in removing class-level information, it remains limited in finer-grained unlearning scenarios, such as instance-wise or attribute-wise unlearning. Exploring source-free unlearning methods tailored to such fine-grained unlearning tasks presents an important direction for future work.

## Ackownledgement

This work was supported in part by the National Natural Science Foundation of China (No.62172362).

## Impact Statement

This paper presents a novel approach to source-free unlearning, advancing the field of machine learning with the proposed DSDA framework, which enhances both efficiency and privacy in unlearning tasks. Our work has the potential to positively impact the field of Machine Learning, particularly in scenarios of data privacy.

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
