# OpenReview forum: "Efficient Source-free Unlearning via Energy-Guided Data Synthesis and Discrimination-Aware Multitask Optimization"
_ICML.cc/2025/Conference — ICML 2025 spotlightposter_

### Official Review · Reviewer_bm1b · 2025-03-09

**Overall Recommendation:** 4

**Summary:**

The authors propose the DSDA source-free unlearning framework the address the challenge of inaccessible original training data. DSDA consists of two key components: (1) AEGDS generates synthetic data using Langevin dynamics, and (2) DAMO balances unlearning objectives by resolving gradient conflicts. Extensive experiments demonstrate that DSDA outperforms existing source-free unlearning methods in terms of both efficacy and efficiency.

**Claims And Evidence:**

The claims made in the submission are supported by clear and convincing evidence

**Essential References Not Discussed:**

The paper thoroughly cite and discuss the related works necessary to understand its key contributions.

**Experimental Designs Or Analyses:**

Yes. The weaknesses are detailed below

**Methods And Evaluation Criteria:**

The proposed method is well-suited to source-free unlearning. The evaluation metrics and datasets are widely used

**Other Comments Or Suggestions:**

n/a

**Other Strengths And Weaknesses:**

Strengths

1.	The authors propose an innovative and practical solution to a pressing and under-explored problem in machine learning.
2.	The findings on feature distribution shifts after unlearning provide compelling evidence for the need for discriminative feature alignment. The observation that traditional unlearning methods disrupt intra-class compactness and inter-class separability highlights a previously overlooked issue.
3.	Extensive experiments on multiple benchmark datasets strongly support the proposed method. Additionally, the ablation studies illustrate the individual contributions of AEGDS, alignment objectives, and multitask optimization.
4.	The paper is well-organized, with clear theoretical foundations and thorough experimental validation.

Weaknesses
1.	The authors claim that the synthetic data closely approximates the original data distribution using feature distribution overlap and visual verification. However, further incorporating quantitative metrics (such as FID) to evaluate the soundness of synthetic data could strengthen the paper.
2.	The framework includes several hyperparameters, especially the weighting factors for multitask optimization, yet the paper provides limited discussion on their potential impact on performance. A deeper analysis of hyperparameter sensitivity would be valuable for understanding the framework’s robustness.

**Questions For Authors:**

Please refer to the weakness

**Relation To Broader Scientific Literature:**

This paper contributes to the under-explored challenge of source-free unlearning, a topic with increasing relevance due to privacy concerns and regulatory demands

**Theoretical Claims:**

Yes. I check the proofs for key theorems (Theorem 3.1 on Langevin dynamics and Theorem 3.3 on multitask gradient optimization)

---

> ### Author Rebuttal · Authors · 2025-03-29
>
> ***W1: Incorporating quantitative metrics (such as FID) to evaluate the soundness of synthetic data could strengthen the paper.***
>
> We respectfully disagree with the reviewer’s suggestion. We emphasize that our goal is to generate synthetic data that approximates the original distribution while preserving privacy. Unlike model inversion methods, which aim to maximize image fidelity, our approach prioritizes unlearning effectiveness without reconstructing identifiable samples. Therefore, evaluating the synthetic data with metrics like FID, which emphasize visual quality, is not necessary for our objectives.
>
> ***W2: Parameter sensitivity experiments w.r.t. weighting factors.***
>
> The weighting factors in our MAMO method are set to ensure that the three losses are on a similar scale. To analyze their sensitivity, we conduct experiments of one-class unlearning task on CIFAR-10. We fix the weight of the retain loss $w_R$ at 0.1 and vary $w_F$ and $w_{Disc}$ within {0.01, 0.1, 0.5, 1}, observing the unlearned model’s $A_r$ and $A_f$. The results in the table below show that when $w_F$ and $w_{Disc}$ are within the range of 0.1–0.5, the model's performance remains stable. However, an excessively large $w_F$ or a small $w_{Disc}$ leads to a noticeable decline in $A_r$.
>
> | $w_R$| $w_F$ | $w_{Disc}$ | $A_r$ | $A_f$ |
> | ---- | ---- | ---- | ---- | ---- |
> | 0.1 | 0.01 | 0.01 | 71.68 | 0.00 |
> | 0.1 | 0.1 | 0.01 | 74.60 | 0.00 |
> | 0.1 | 0.5 | 0.01 | 74.23 | 0.00 |
> | 0.1 | 1 | 0.01 | 73.14 | 0.00 |
> | 0.1 | 0.01 | 0.1 | 76.92 | 0.00 |
> | 0.1 | 0.1 | 0.1 | 79.25 | 0.00 |
> | 0.1 | 0.5 | 0.1 | 78.07 | 0.00 |
> | 0.1 | 1 | 0.1 | 77.03 | 0.00 |
> | 0.1 | 0.01 | 0.5 | 74.38 | 0.00 |
> | 0.1 | 0.1 | 0.5 | 78.35 | 0.00 |
> | 0.1 | 0.5 | 0.5 | 78.05 | 0.00 |
> | 0.1 | 1 | 0.5 | 77.52 | 0.00 |
> | 0.1 | 0.01 | 1 | 72.46 | 0.00 |
> | 0.1 | 0.1 | 1 | 77.67 | 0.00 |
> | 0.1 | 0.5 | 1 | 78.02 | 0.00 |
> | 0.1 | 1 | 1 | 77.26 | 0.00 |

---

### Official Review · Reviewer_NZ7b · 2025-03-11

**Overall Recommendation:** 4

**Summary:**

This paper proposes a new framework, DSDA, for machine unlearning without access to the training data.

Specifically, DSDA first crafts synthetic data via energy-based models with Langevin dynamics and then performs unlearning using the generated synthetic data.

The authors observe that simply formulating the unlearning problem as $\lambda\_1 \mathcal{L}\_R + \lambda\_2 \mathcal{L}\_F$ would let retain class samples become dispersed and disrupt feature distributions; therefore, DSDA include a feature alignment objective $\mathcal{L}\_{Disc}$ to improve intra-class compactness and inter-class separability of retain classes.

Furthermore, since the objective now involves $\mathcal{L}\_R, \mathcal{L}\_F$ and $\mathcal{L}\_{Disc}$, gradient conflicts may happen during the unlearning process. Hence, DSAD employ a multitask optimization strategy to make sure the update vector is close to the joint gradient.

Experiments on CIFAR-10, CIFAR-100, and PinsFaceRecognition datasets across CNN and ViT demonstrate the effectiveness of the proposed framework DSDA compared to baselines.

**Claims And Evidence:**

In general, the claims made in the submission are clear and convincing.

However, regarding the observation that simply formulating the unlearning problem as $\lambda\_1 \mathcal{L}\_R + \lambda\_2 \mathcal{L}\_F$ would let retain class samples become dispersed and disrupt feature distributions, I recommend the authors provide more evidence to support that.

In the submission, an empirical result on CIFAR-10 when forgetting class 1 is provided. However, $\lambda\_1, \lambda\_2$ may affect the results, as well as the forgotten class, \i.e., for scenarios where classes having hierarchical categories may affect the distributions and be more difficult.

**Essential References Not Discussed:**

N/A

**Experimental Designs Or Analyses:**

The experimental setting is suitable and correct.

Please refer to Claims And Evidence* for the concerns on the analysis of observations.

**Methods And Evaluation Criteria:**

Yes, the proposed framework DSDA is important to improve the practicality of machine unlearning as training data may not be available, and DSDA not need to collect auxiliary data.

**Other Comments Or Suggestions:**

- It would be better to explain why not adopting contrastive alignment [1-2] for the discriminative feature alignment objective, as the former can also obtain the objective and can drop weight factors $\alpha$.

- [3] discusses the gradient conflict issue in machine unlearning and employs an optimization method to resolve the issue. It would be better to explain the rationale behind the choice of the solution to gradient conflicts in the submission.

- It would be better to admit and discuss limitations such as class-wise forgetting in the paper.


I am willing to raise my score if the authors can address my main concerns.

----
> [1] Liu, Qingxiang, et al. "Personalized Federated Learning for Spatio-Temporal Forecasting: A Dual Semantic Alignment-Based Contrastive Approach." arXiv preprint arXiv:2404.03702 (2024).
>
> [2] Tan, Yue, et al. "Is heterogeneity notorious? Taming heterogeneity to handle test-time shift in federated learning." Advances in Neural Information Processing Systems 36 (2023): 27167-27180.
>
> [3] Wu, Jing, et al. "Erasediff: Erasing data influence in diffusion models." arXiv preprint arXiv:2401.05779 (2024).

**Other Strengths And Weaknesses:**

The proposed DSDA crafts synthetic data via energy-based models with Langevin dynamics, providing a fresh perspective on source-free machine unlearning. However, this mechanism may be limited on high-resolution data.
In addition, the proposed framework is limited to class-wise forgetting.

**Questions For Authors:**

- Regarding the objective $\mathcal{L}_F$, what if DSDA apply random labelling? Would it help with feature alignment?

- What is the data range for the crafted data?

**Relation To Broader Scientific Literature:**

Compared to existing source-free machine unlearning methods, the key contribution of this work is that there is no need to collect auxiliary data and no need to train an auxiliary model/model retraining.

**Theoretical Claims:**

The proof seems correct.

---

> ### Author Rebuttal · Authors · 2025-03-29
>
> ***W1: This mechanism may be limited on high-resolution data.***
>
> Thank you for the comment. However, we respectfully disagree. Our method is not restricted to high-resolution data, as demonstrated by our experiments on three datasets with varying resolutions—CIFAR-10 (32×32), CIFAR-100 (32×32), and PinsFaceRecognition (224×224). The consistent performance across both low- and high-resolution scenarios validates the broad applicability of our method.
>
> ***W2&C3: The proposed framework is limited to class-wise forgetting.***
>
> We appreciate the reviewer’s insight. Our method is designed for class-wise unlearning, as it aligns with common unlearning scenarios and is more practical in real-world applications. Most existing unlearning studies also focus on class-level or concept-level unlearning, as instance-level unlearning poses inherent challenges—removing a specific instance does not guarantee that the model will completely forget similar patterns, as retain data with similar features may still contribute to strong performance on the forgot data. While our method is effective in removing class-level information, we recognize its limitations in finer-grained unlearning, such as instance-wise or attribute-wise unlearning, which we will clarify in our final version.
>
> ***C1: Why not adopting contrastive alignment for the discriminative feature alignment objective?***
>
> The contrastive alignment methods in [1-2] employ a metric learning loss that is fundamentally similar to our $L_{disc}$ without the weight factors $\alpha$. However, our method introduces $\alpha$ to adaptively adjust gradient magnitudes based on how far a similarity score deviates from its optimal values.
> This adaptive weighting mechanism provides two key advantages. First, it prioritizes large updates for highly misaligned features, accelerating convergence and improving optimization efficiency. Second, it prevents excessive penalization of already well-aligned features, reducing the risk of overcorrection. As a result, our method achieves more precise and stable discriminative feature alignment, ultimately enhancing unlearning effectiveness.
>
> ***C2: Explain rationale behind the choice of the solution to gradient conflicts.***
>
> The method in [3] primarily balances two losses and requires a predefined distinction between the main and auxiliary objectives, limiting its applicability to more complex multi-objective optimization. In contrast, our approach can effectively balance three or more objectives without the need for a manually designated main objective, enabling a more flexible and adaptive optimization process. This allows for the seamless integration of the Discriminative Feature Alignment Objective and ensures a better trade-off among multiple competing objectives in unlearning.
>
> ***Q1: Regarding the objective LF, what if DSDA apply random labelling? Would it help with feature alignment?***
>
> We appreciate the reviewer’s thought-provoking question. However, random labeling may not be an effective alternative for the forget objective $L_F$. As shown in Figure 2(b), most forget data samples are predicted as specific classes rather than randomly distributed across all classes. Additionally, prior research [1] has shown that fine-tuning a trained model with randomly labeled forget data can introduce unintended shifts in the decision boundaries of retain classes, ultimately degrading the model’s utility on the retain data. Therefore, the forget objective  $L_F$, which explicitly optimizes unlearning through gradient ascent, is more controlled and effective in achieving the desired unlearning behavior.
>
> [1] Chen, Min, et al. "Boundary unlearning: Rapid forgetting of deep networks via shifting the decision boundary." Proceedings of the IEEE/CVF Conference on Computer Vision and Pattern Recognition. 2023.
>
> ***Q2: What is the data range for the crafted data?***
>
> The crafted data follows a similar range to the preprocessed original data. For example, in the case of CIFAR-10, we apply a common normalization setting with CIFAR_MEAN = (0.5071, 0.4865, 0.4409) and CIFAR_STD = (0.2673, 0.2564, 0.2762), resulting in a transformed data range of approximately [−2,2]. Our synthetic data is generated within a slightly extended range of [−2.5,2.5], ensuring compatibility with the model while maintaining sufficient variability for effective unlearning.

---

> > ### Comment · Reviewer_NZ7b · 2025-04-06
> >
> > Thanks for the response. The authors addressed my concerns, so I increased my final score to 4.

---

### Official Review · Reviewer_nSY3 · 2025-03-11

**Overall Recommendation:** 4

**Summary:**

The paper addresses the challenge of source-free unlearning for image classification ML models, where the training data cannot be accessed after initial model training. The paper proposes a novel framework called DSDA, which utilizes Langevin dynamics, Runge–Kutta methods and gradient-based multitask optimization to achieve source-free unlearning. The paper demonstrate that DSDA achieves superior efficiency and effectiveness through experiments on three datasets. The paper also conducts ablation studies for key components and presents visualization analysis of the synthetic data.

**Claims And Evidence:**

The authors provide adequate theoretical and experimental evidence for their claims.

**Essential References Not Discussed:**

The authors well discussed existing works on machine unlearning.

**Experimental Designs Or Analyses:**

Yes. The experimental design is adequate, including comparisons with sota methods, ablation studies and visualizations.

**Methods And Evaluation Criteria:**

The proposed method and evaluation criteria are well-suited and relevant for the problem.

**Other Comments Or Suggestions:**

See Weaknesses for details.

**Other Strengths And Weaknesses:**

Strengths :

1.	The paper addresses a practical and critical problem of source-free unlearning. Unlike existing methods that either require access to training data or incur high computational costs, the proposed framework introduces an efficient alternative by leveraging energy-guided data synthesis and multitask optimization.

2.	The incorporation of Langevin dynamics-based sampling for synthetic data generation is an innovative approach that bridges model inversion, generative modeling, and unlearning. With the proposed AEGDS method, DSDA directly reconstructs data distributions without external generators, enhancing both privacy protection and computational efficiency.

3.	The paper includes extensive experiments across multiple datasets and compare DSDA with sota baselines, demonstrating its superiority in both unlearning effectiveness and efficiency.

Weaknesses :

1.	A more detailed discussion on the potential limitations of the synthetic data (e.g., its ability to generalize to different data distributions) would enhance the robustness of the argument.

2.	The legend in Figure 5 (a) appears to be small, which may affect readability. Additionally, the color contrast could be improved to enhance clarity.

**Questions For Authors:**

Please response to the weaknesses above.

**Relation To Broader Scientific Literature:**

The paper contributes to the emerging field of source-free unlearning, which aims to remove specific data influence from a trained model without access to the original training dataset. The findings on feature distribution changes after unlearning contribute to the growing body of work on explainable AI and model interpretability.

**Theoretical Claims:**

Yes. The theoretical claims regarding the AEGDS and DAMO are well-supported.

---

> ### Author Rebuttal · Authors · 2025-03-29
>
> ***W1: A deeper discussion on the synthetic data's limitations would strengthen the argument.***
>
> We appreciate the reviewer’s suggestion. Our experiments on three datasets with 10, 100, and 105 classes demonstrate that the synthetic data effectively supports unlearning across diverse distributions. As shown in Section 4.4, the visualized feature distributions further confirm that the generated data closely approximates real data across various classes. However, we acknowledge that for certain distributions, such as highly sparse or imbalanced data, our method may not fully capture fine-grained structural details. Addressing this limitation is an important direction for future research.
>
> ***W2: The legend in Figure 5 (a) appears to be small, which may affect readability. Additionally, the color contrast could be improved to enhance clarity.***
>
> We thank the reviewer for pointing out this issue. We will optimize Figure 5 (a) in our final version.

---

### Official Review · Reviewer_gZDa · 2025-03-11

**Overall Recommendation:** 4

**Summary:**

The authors present a well-structured and innovative approach to source-free unlearning, where a trained model must forget specific data without access to the original training dataset. To achieve this, the authors propose a novel two-stage framework DSDA. Firstly, the proposed AEGDS generates synthetic dataset as a substitute for the original ones using Langevin dynamics, enhanced with Runge–Kutta and momentum-based acceleration. Secondly, based on findings of the unlearned feature distribution, the authors introduce a novel discrimination-aware unlearning objective and perform balanced optimization to achieve unlearning. The authors conduct adequate experiments on three datasets and multiple unlearning tasks. Results show that DSDA outperforms existing source-free methods and is comparable to general methods, in terms of efficiency and effectiveness.

**Claims And Evidence:**

Yes, the claims are well supported.

**Essential References Not Discussed:**

There are no related works not discussed.

**Experimental Designs Or Analyses:**

Yes. Detailed in “Other strengths and weaknesses”.

**Methods And Evaluation Criteria:**

The proposed methods and evaluation criteria make sense for the problem or application at hand.

**Other Comments Or Suggestions:**

N/A.

**Other Strengths And Weaknesses:**

Strengths :

1.	The authors presents and addresses the pressing problem of source-free unlearning, a critical challenge in real-world machine unlearning scenarios where access to training data is restricted. The proposed framework offers a novel and efficient solution, surpassing existing source-free methods that rely on knowledge distillation.
2.	The authors provides insightful visualizations of feature space to illustrate the impact of unlearning, highlighting how traditional methods disrupt feature distributions. These insights empirically justify the need for discriminative feature alignment, strengthening the theoretical motivation.
3.	The metrics in the experiment are comprehensive, including accuracy, efficiency and defensive capability.
4.	The authors conduct ablation experiments to isolate the contributions of each component of DSDA, clearly demonstrating the importance of each part of the framework in achieving the overall performance improvements.
5.	The writing is clear and well-structured. In particular, the authors clearly introduce the core components AEGDS and DAMO with intuitive figures, algorithmic pseudocode, and detailed explanations, ensuring that the methodology is easy to understand and implement.

Weaknesses :

1.	Because the METHOD is written with many symbols and equations, adding a notation table will make the paper clearer.

**Questions For Authors:**

1. As the proposed AEGDS generates synthetic data that approximates the original training distribution, I am concerned that could the synthetic data cause potential privacy linkage?
2. The authors mention weighting factors for multitask optimization, but provide limited discussion on their impact. Could they provide insights or experiments showing how these factors affect the framework’s performance and balance between unlearning objectives?

**Relation To Broader Scientific Literature:**

The paper also contribute to the fields of model inversion and multitask learning.

**Theoretical Claims:**

Yes (e.g. theorem 3.1 and 3.3).

---

> ### Author Rebuttal · Authors · 2025-03-29
>
> ***W1: Adding a notation table will make the paper clearer.***
>
> We thank the reviewer for pointing out this issue. We will add a notation table in our final version.
>
> ***Q1: Could the synthetic data cause potential privacy linkage?***
>
> The reviewer raises a critical concern. However, our work addresses this concern through an empirical analysis presented in Section 4.4. Specifically, we visualize the generated synthetic samples and demonstrate that they are visually indistinguishable from random noise, making it impossible for human observers to extract any meaningful information. This observation confirms that the synthetic data does not retain identifiable features of the original training data, thereby mitigating potential privacy risks.
>
> ***Q2: Parameter sensitivity experiments w.r.t. weighting factors.***
>
> The weighting factors in our MAMO method are set to ensure that the three losses are on a similar scale. To analyze their sensitivity, we conduct experiments of one-class unlearning task on CIFAR-10. We fix the weight of the retain loss $w_R$ at 0.1 and vary $w_F$ and $w_{Disc}$ within {0.01, 0.1, 0.5, 1}, observing the unlearned model’s $A_r$ and $A_f$. The results in the table below show that when $w_F$ and $w_{Disc}$ are within the range of 0.1–0.5, the model's performance remains stable. However, an excessively large $w_F$ or a small $w_{Disc}$ leads to a noticeable decline in $A_r$.
>
> | $w_R$| $w_F$ | $w_{Disc}$ | $A_r$ | $A_f$ |
> | ---- | ---- | ---- | ---- | ---- |
> | 0.1 | 0.01 | 0.01 | 71.68 | 0.00 |
> | 0.1 | 0.1 | 0.01 | 74.60 | 0.00 |
> | 0.1 | 0.5 | 0.01 | 74.23 | 0.00 |
> | 0.1 | 1 | 0.01 | 73.14 | 0.00 |
> | 0.1 | 0.01 | 0.1 | 76.92 | 0.00 |
> | 0.1 | 0.1 | 0.1 | 79.25 | 0.00 |
> | 0.1 | 0.5 | 0.1 | 78.07 | 0.00 |
> | 0.1 | 1 | 0.1 | 77.03 | 0.00 |
> | 0.1 | 0.01 | 0.5 | 74.38 | 0.00 |
> | 0.1 | 0.1 | 0.5 | 78.35 | 0.00 |
> | 0.1 | 0.5 | 0.5 | 78.05 | 0.00 |
> | 0.1 | 1 | 0.5 | 77.52 | 0.00 |
> | 0.1 | 0.01 | 1 | 72.46 | 0.00 |
> | 0.1 | 0.1 | 1 | 77.67 | 0.00 |
> | 0.1 | 0.5 | 1 | 78.02 | 0.00 |
> | 0.1 | 1 | 1 | 77.26 | 0.00 |

---

### Decision · Program_Chairs · 2025-05-01

**Decision:**

Accept (spotlight poster)

**Comment:**

Hi,

Draft has received overall positive reviews, with 4 accept.  Dear authors, please update the draft per recommendations and comments of the reviewers.

Congratulations.

regards

AC